# Physiological and Genomic Characterization of a Novel Obligately Chemolithoautotrophic, Sulfur-Oxidizing Bacterium of Genus *Thiomicrorhabdus* Isolated from a Coastal Sediment

**DOI:** 10.3390/microorganisms11102569

**Published:** 2023-10-15

**Authors:** Yu Gao, Han Zhu, Jun Wang, Zongze Shao, Shiping Wei, Ruicheng Wang, Ruolin Cheng, Lijing Jiang

**Affiliations:** 1Key Laboratory of Marine Genetic Resources, Third Institute of Oceanography, Ministry of Natural Resources, Xiamen 361005, China22320201151293@stu.xmu.edu.cn (J.W.); nmseisvsiecrsan@gmail.com (R.W.); 2State Key Laboratory Breeding Base of Marine Genetic Resources, Xiamen 361005, China; 3School of Marine Sciences, China University of Geosciences, Beijing 100083, China; 4School of Marine Biology, Xiamen Ocean Vocational College, Xiamen 361005, China

**Keywords:** *Thiomicrorhabdus*, sulfur-oxidation, extracellular sulfur, coastal sediment, chemolithoautotroph

## Abstract

*Thiomicrorhabdus* species, belonging to the family *Piscirickettsiaceae* in the phylum *Pseudomonadotav* are usually detected in various sulfur-rich marine environments. However, only a few bacteria of *Thiomicrorhabdus* have been isolated, and their ecological roles and environmental adaptations still require further understanding. Here, we report a novel strain, XGS-01^T^, isolated from a coastal sediment, which belongs to genus *Thiomicrorhabdus* and is most closely related to *Thiomicrorhabdus hydrogeniphila* MAS2^T^, with a sequence similarity of 97.8%. Phenotypic characterization showed that XGS-01^T^ is a mesophilic, sulfur-oxidizing, obligate chemolithoautotrophy, with carbon dioxide as its sole carbon source and oxygen as its sole electron acceptor. During thiosulfate oxidation, strain XGS-01^T^ can produce extracellular sulfur of elemental α-S_8_, as confirmed via scanning electron microscopy and Raman spectromicroscopy. Polyphasic taxonomy results indicate that strain XGS-01^T^ represents a novel species of the genus *Thiomicrorhabdus*, named *Thiomicrorhabdus lithotrophica* sp. nov. Genomic analysis confirmed that XGS-01^T^ performed thiosulfate oxidation through a *sox* multienzyme complex, and harbored *fcc* and *sqr* genes for sulfide oxidation. Comparative genomics analysis among five available genomes from *Thiomicrorhabdus* species revealed that carbon fixation via the oxidation of reduced-sulfur compounds coupled with oxygen reduction is conserved metabolic pathways among members of genus *Thiomicrorhabdus*.

## 1. Introduction

The genus *Thiomicrorhabdus* belonging to the family *Piscirickettsiaceae* of the phylum *Pseudomonadota* (formerly named Proteobacteria) was first proposed based on phylogenetic analyses, including the 16S rRNA gene and 53 ribosomal protein sequences, by Boden et al. [1]. Members of genus *Thiomicrorhabdus* were distributed in various habitats, including deep-sea hydrothermal vents, continental shelf sediments, and intertidal mud flats [2,3,4,5]. *Thiomicrorhabdus* species are strict chemolithoautotrophs and have diverse energy metabolic pathways [6,7]. They generally oxidize reduced sulfur compounds to fix CO_2_ with oxygen or nitrate as electron acceptors and synthesize the primary productivity of ecosystems in the process [2,3,5].

At the time of writing, the genus *Thiomicrorhabdus* has 12 reported species with validly published names. *Thiomicrorhabdus frisia* JB-A2^T^ is the first validly published species for the genus *Thiomicrorhabdus* and is isolated from marine sediments along the Wadden Sea [5]. Subsequently, *Thiomicrorhabdus chilensis* Ch-1^T^ was isolated from the coastal shelf of Chilean [2]. In 2005, Knittel et al. identified the first cryophilic sulfur-oxidizing bacteria of this genus, *Thiomicrorhabdus psychrophile* SVAL-D^T^ and *Thiomicrorhabdus arctica* SVAL-E^T^ [3]. *Thiomicrorhabdus hydrogeniphila* MAS2^T^ was isolated from a tank containing coastal seawater from Tokyo Bay [8]. *Thiomicrorhabdus aquaedulcis* HaS4^T^ was isolated from the water of Lake Harutori in Japan [9]. *Thiomicrorhabdus indica* 13–15A^T^ were isolated from the deep-sea hydrothermal vent environment of Southwest Indian Ocean [6]. *Thiomicrorhabdus sediminis* G1^T^ and *Thiomicrorhabdus xiamenensis* G2^T^ were isolated from sediment samples collected from the coast of Xiamen [7]. Recently, *Thiomicrorhabdus heinhorstiae* [10], *Thiomicrorhabdus cannonii* [10], and *Thiomicrorhabdus immobilis* [11] were discovered one after another. 

Although genus *Thiomicrorhabdus* have been detected from worldwide marine systems, its physiological diversity and ecological role have not yet been made fully explicit due to the lack of cultivated strains. In this study, we isolated a new strain, named XGS-01^T^, from the coastal sediment of Xiamen, China. We characterized the bacterium as a novel species of genus *Thiomicrorhabdus* using a polyphasic taxonomic approach; meanwhile, we compared genomic analyses among members of the genus *Thiomicrorhabdus* to gain insights into the ecological roles and environmental adaptation mechanisms underlying their widespread distribution. Furthermore, we determined the characteristics of extracellular sulfur produced by *Thiomicrorhabdus* bacteria. 

## 2. Materials and Methods

### 2.1. Enrichment and Isolation

Samples were collected in February 2021 from coastal sediments at Xiamen, Fujian, China (118°12′ E, 24°29′ N), and kept cold in dark during transportation to the laboratory. Samples were stored at 4 °C before further processing. A total of 1 g of sediment samples were transferred to 10 mL of modified MMJS medium [12] containing NaCl (15 g/L), CaCl_2_·2H_2_O (0.14 g/L), NH_4_Cl (0.25 g/L), MgCl_2_·6H_2_O (4.18 g/L), MgSO_4_·7H_2_O (3.4 g/L), KCl (0.33 g/L), K_2_HPO_4_ (0.14 g/L), NiCl_2_·6H_2_O (0.5 mg/L), Fe(NH_4_)_2_(SO_4_)_2_·6H_2_O (0.01 g/L), Na_2_SeO_3_·5H_2_O (0.5 mg/L), NaHCO_3_ (1 g/L), Wolfe trace elements 10 mL/L, and Wolfe vitamin 1 mL/L [13]. Before autoclaving, materials, except the vitamin solution, Na_2_S_2_O_3_·5H_2_O and NaHCO_3_, were dissolved, and the pH value of the medium was adjusted to 7.0. After autoclaving, the filter-sterilized vitamin solution, Na_2_S_2_O_3_·5H_2_O and NaHCO_3_, were added into the medium. Then, 10 mL medium was anaerobically dispensed into 50 mL serum bottles with gas purging and sealed with a butyl-rubber stopper under a gas phase of 76% N_2_/20% CO_2_/4% O_2_ (200 KPa) [14]. After 3 days of incubation at 28 °C, the cell growth was determined under the light microscope (Eclipse 80i, Nikon, Tokyo, Japan). Strain XGS-01^T^ was obtained from the highest dilution via the dilution-to-extinction technique, and the purity was confirmed via microscopic examination and 16S rRNA gene sequencing.

### 2.2. DNA Extraction and Genomic Analyses

The genomic DNA of strain XGS-01^T^ was extracted according to a previously reported method [15]. The complete genome of strain XGS-01^T^ was sequenced using SMRT technology performed by Shanghai Majorbio Bio-pharm Technology Co., Ltd. (Shanghai, China). The sequencing was performed on a Pacific Biosciences (PacBio) sequencing platform via single-molecule real-time (SMRT) technology. The sequenced reads were filtered, and high-quality paired-end reads were assembled to reconstruct the circulating genome by using SOAPdenovo2 software. The G+C content of chromosomal DNA was determined from the complete genome sequences. 

### 2.3. 16S rRNA Gene Phylogeny

The 16S rRNA gene was PCR-amplified and sequenced with 27F (5′-AGAGTTGATCMTGCGCTCAG-3′) and 1492R (3′-TACGYTACGTCTTACGACT-5′) primers. The almost full-length 16S rRNA gene sequence of strain XGS-01^T^ was analyzed using the gapped-blast search algorithm [16]. The gene sequences of the strains were edited by DNAMAN and used to identify the 16S rRNA sequence similarities and phylogenetic neighbors by NCBI (https://www.ncbi.nlm.nih.gov/, accessed on 14 May 2023) and the Ez Biocloud server (https://www.ezbiocloud.net, accessed on 21 May 2023) [17]. The 16S rRNA sequences with taxa closely related to the strain XGS-01^T^ were selected for downloading. The phylogenetic tree was constructed using MEGA 11.0 software [18] with three methods—the neighbor-joining (NJ), maximum-likelihood (ML), and maximum-parsimony (MP) methods [19]—for the target sequences and all downloaded gene sequences. Genetic distance was calculated using the Kimura two-parameter model [20]. Bootstrap analysis was calculated based on 1000 replications.

### 2.4. Morphology, Physiology, and Chemotaxonomic Analysis

Morphological observations of the novel isolated XGS-01^T^ was performed via a transmission electron microscopy (JEM-1400, JEOL, Tokyo, Japan) with cultures grown in MMJS liquid medium at 32 °C for 1 day. The Gram stain test was conducted with a Gram-staining kit (Hangzhou Tianhe Microbiological Reagent Co., Ltd., Hangzhou, China).

The growth of strain XGS-01^T^ was measured via direct cell counting using a phase contrast microscope (Eclipse 80i, Nikon). Triplicate cultures were examined under each condition, and cultures were grown in MMJS medium to determine the physiological characterization of strain XGS-01^T^. The growth temperature was tested at 0, 4, 10, 15, 20, 25, 28, 30, 35, 37, 45, and 55 °C. The pH range for growth was determined as from pH 4.5 to 9.0 at increments of 0.5 pH units using 30 mM hydrogen phosphate–citrate buffer (pH 4–5), MES (pH 5–6), PIPES (pH 6–7), HEPES (pH 7–7.5), and Tris (pH 8–9.5) at 30 °C. The salt concentration for growth was investigated by using MMJS medium supplemented with 0, 170, 340, 510, 680, 850, 1020, 1190, 1360, 1530, and 1700 mM NaCl. The effect of O_2_ on growth was measured at 30 °C by setting different O_2_ content gradients (0, 1, 2, 4, 6, 8, and 10% at 200 kPa and 21% at 100 kPa), and 10 mM nitrate was added as a potential electron acceptor for oxygen absence condition. After determining the optimal growth conditions, subsequent cultivation will use the optimal growth conditions.

The potential electron donors were investigated using sulfite (5 mM), thiocyanate (5 mM), tetrathionate (5 mM), elemental sulfur (1% *w*/*v*), or sodium sulfide (50 and 100 μM) to replace thiosulfate in MMJS medium, and the gas phase is 76% N_2_/20% CO_2_/4% O_2_ (200 kPa). The utilization of hydrogen was also examined with 76% H_2_/20% CO_2_/4% O_2_ (200 kPa) in the absence of thiosulfate. As to other electron acceptors, nitric acid (0.1%, *w*/*v*), nitrite (0.1 and 0.01%, *w*/*v*), sulfite (0.01%, *w*/*v*), and elemental sulfur (1%, *w*/*v*) were tested in MMJS medium with sodium thiosulfate as the electron donor. Heterotrophic growth was examined in MMJS medium by replacing NaHCO_3_ with potential organic carbon sources under the gas phase of 98% N_2_/2% O_2_ (200 kPa). The organic carbon sources including 20 amino acids (methionine, glutamine, glycine, alanine, valine, lysine, arginine, leucine, isoleucine, phenylalanine, proline, tyrosine, cysteine, threonine, tryptophan, serine, aspartic acid, and histidine) at a final concentration of 5 mM; 0.02% (*w*/*v*) glucose, sucrose, galactose, lactose, fructose, maltose and alginate; 5 mM formate, acetate, propionate, citrate, succinate, pyruvate; 0.1% (*w*/*v*) peptone, yeast extract, tryptone, starch; and a gas phase of 96% N_2_/4% O_2_ (200 kPa). In an attempt to determine the alternative energy source, these organic compounds were used as an energy source to replace thiosulfate in MMJS medium with the gas phase of 78% N_2_/20% CO_2_/2% O_2_ (200 kPa). To detect the utilization of nitrogen sources for XGS-01^T^, NH_4_Cl (5 mM), NaNO_3_ (5 mM), NaNO_2_ (5 mM), nitrilotriacetic acid (5 mM), and N_2_ were added to the MMJS medium without NH_4_Cl. The gas phase was 78% H_2_/20% CO_2_/4% O_2_ (200 kPa) and 78% N_2_/20% CO_2_/4% O_2_ (200 kPa). To detect antibiotic resistance, bacterial growth was detected by adding 0.1 g/L kanamycin, rifampicin, chloramphenicol, ampicillin, and streptomycin in MMJS medium, and other conditions were the same as the culture conditions. 

Fatty acids of whole cells grown in the MMJS medium at 28 °C for 24 h were saponified, methylated, and extracted using the standard midi protocol (Sherlock Microbial Identification System, version 6.0B). Using the same method, the cellular fatty acid composition of *T. sediminis* G1^T^ and *T. xiamenensis* G2^T^ was also determined in parallel with that of strain XGS-01^T^ in this study. 

### 2.5. The Determination of Extracellular Sulfur via Scanning Electron Microscopy and Raman Spectromicroscopy

In order to determine the characteristics of extracellular sulfur produced by strain XGS-01^T^, strain XGS-01^T^ was incubated at 32 °C for 12 h, and subsequently, the extracellular sulfur was collected by using 0.2 μm filter membrane and rinsed it with phosphate buffer solution (PBS) before centrifugation, repeating three times to remove residual culture medium on the surface of elemental sulfur. The collected elemental sulfur sample was determined using scanning electron microscopy (SEM) (ZEISS Sigma FE-SEM, Carl Zeiss, Jena, Germany), Raman spectromicroscopy (Horiba LabRAM HR-evolution, Longjumeau, France), and X-ray diffraction analysis (HyPix-400, Rigaku Ultima IV, Tokyo, Japan).

## 3. Results and Discussion

### 3.1. Phylogenetic Analysis Based on 16S rRNA Gene

The phylogenetic tree was constructed based on the NJ method for the complete 16S rRNA gene sequence (1553 bp) of XGS-01^T^, showing that strain XGS-01^T^ was clustered with *Thiomicrorhabdus* species (Figure 1). The phylogenetic tree based on the ML method (Appendix A) and the MP method (Appendix A) also supports this topology. These results indicated that strain XGS-01^T^ belonged to genus *Thiomicrorhabdus.* BLAST comparison showed that strain XGS-01^T^ had the highest sequence similarity with *T. hydrogeniphila* MAS2^T^ [21], with 97.8% similarity, followed by *T. psychrophila* SVAL-D^T^ (97.4% similarity) [3] and *T. frisia* JB-A2^T^ (97.3% similarity) [5]. This value is lower than the threshold criterion for prokaryotic species delineation, which is 98.65% for 16S rRNA gene similarity, indicating that strain XGS-01^T^ represents a potential new species within the genus *Thiomicrorhabdus.*

### 3.2. Morphology, Physiology, and Chemotaxonomic Analysis

Strain XGS-01^T^ was rod-shaped, about 0.7–0.8 μm wide and 1.6–1.8 μm long, and had non-flagella (Figure 2), which was similar to other strains in the genus *Thiomicrorhabdus*. Cells were Gram stain negative. Strain XGS-01^T^ could utilize a variety of reduced sulfur compounds as electron donors, including thiosulfate, sulfide, sulfur, and tetrathionate, but not sulfite. The strain XGS-01^T^ cannot use hydrogen as an electron donor, which differs from that of its closely related species strain MAS2^T^. Oxygen is the sole electron acceptor for strain XGS-01^T^. In addition, strain XGS-01^T^ could not utilize all tested organic matters as carbon sources and energy sources. These results indicate that strain XGS-01^T^ is an obligate chemolithoautotrophic sulfur-oxidizing bacterium (Table 1). For the utilization of nitrogen sources, strain XGS-01^T^ is able to utilize both inorganic nitrogen sources, such as ammonium, nitrogen and nitrate, and organic nitrogen, such as aminotriacetic acid. Strain XGS-01^T^ could not use nitrite as the nitrogen source. The result is significantly different from strain MAS2^T^, which utilizes ammonium as the nitrogen source. Kanamycin, rifampicin, chloramphenicol, ampicillin, and streptomycin could inhibit the growth of strain XGS-01^T^, which was basically consistent with other strain types of this genus. The major fatty acids of strain XGS-01^T^ are C_18:1_ (50.14%), C_16:0_ (15.97%), and C_16:1_ (6.90%). There are significant differences between strain XGS-01^T^ and the closely related species, *T. hydrogeniphila* MAS2^T^. The proportion of fatty acid C_16:1_ in strain XGS-01^T^ is 6.90%, whereas it is the main fatty acid in strain MAS2^T^, accounting for 44.40%. In addition, some fatty acids detected in strain XGS-01^T^ were not found in strain MAS2^T^. The profiles of fatty acids in strain XGS-01^T^ were also different from those of strains G1^T^ and G2^T^, such as C_18:0_ (Table 2).

### 3.3. Genome Features and Central Metabolism among Genus Thiomicrorhabdus

The genome size of strain XGS-01^T^ was 2.5 Mb, and the G+C content of strain XGS-01^T^ was 39.1 mol%, which is similar to those of *T. hydrogeniphila* MAS2^T^ (39.6 mol%) [8] and *T*. *frisia.* JB-A2^T^ (39.6 mol%) [5] and lower than that of *T. chilensis* DSM 12352^T^ (49.9 mol%) [2]. The average nucleotide identity (ANI) values between strain XGS-01^T^ and other species were below the thresholds for species delineation (95–96%) [22], indicating that strain XGS-01^T^ represents a novel species of the genus *Thiomicrorhabdus*. None of them have plasmids.

Central metabolic pathways, including sulfur oxidation, nitrogen, and carbon metabolism, were further analyzed among members of genus *Thiomicrorhabdus*, including the complete genome of strain XGS-01^T^ obtained in this study, and *T. frisia* Kp2^T^, *T. arctica* DSM 13458^T^, *T. chilensis* DSM 12352^T^, and *Thiomicrorhabdus* sp. Milos-T2^T^, genomes of which are currently available in the NCBI database. The genome of strain XGS-01^T^ contains a complete set of *soxABCDXYZ* genes encoding the Sox multienzyme complex, which performs thiosulfate oxidation. The gene cluster is ubiquitously present in all species of the genus *Thiomicrorhabdus.* This result shows that thiosulfate oxidation performed via a Sox multienzyme complex is highly conserved among *Thiomicrorhabdus* species. Strain XGS-01^T^ has *sqr* and *fcc* genes for sulfide oxidation, which were also found in other species, except *T. arctica* SVAL-E^T^, which lacks the *fcc* gene (Table 3). There are no complete sulfate-reducing genes and *dsr* genes in the genome of strain XGS-01^T^. All genes encoding the carboxyl oxygenase (Rubisco) and ribulose diphosphate (RuBP), essential for carbon fixation, are present in the genome of strain XGS-01^T^. In addition, strain XGS-01^T^ possesses two carbonic anhydrases, including α-type and γ-type. There are some differences in carbonic anhydrase among these species; for example, *T. frisia* JB-A2^T^, *T. arctica* DSM 134585^T^, and *Thiomicrorhabdus* sp. Milos-T2^T^ have α-type, β-type, and γ-type, whereas *T. chilensis* DSM 12352^T^ has α-type and γ-type carbonic anhydrase. Strain XGS-01^T^ has an incomplete nitrogen metabolism pathway. The genome has no genes involved in the denitrification pathway. The genes encoding nitrate reductase and nitrite reductase are absent in the genome of strain XGS-01^T^. The incomplete nitrogen metabolism pathway was also found in the other species, except *T. frisia* JB-A2^T^.

### 3.4. Characterization of Extracellular Sulfur Produced by Thiomicrorhabdus Species with SEM, XRD, and Raman Spectroscopy

Elemental sulfur was present in the culture during the mid-exponential growth phase when strain XGS-01^T^ was incubated with thiosulfate and oxygen. Previous studies showed that the accumulation of elemental sulfur as an intermediate metabolite of thiosulfate oxidation only occurred in microorganisms lacking both SoxCD and *dsr* operons [23,24,25]. In this study, strain XGS-01^T^ can not only oxidize thiosulfate to produce sulfate through the Sox system; it can also secrete extracellular sulfur during this process despite the presence of SoxCD genes in its genomes. The reason for extracellular sulfur production may be the inhibition of SoxCD. The expression of SoxCD usually depends on environmental pH, and the optimal pH value for the sulfite dehydrogenase (SoxCD) activity is 8.0 [26]. During the thiosulfate oxidation process of strain XGS-01^T^, the pH of medium will gradually decrease due to the production of metabolites, such as sulfate, and then inhibit the expression of SoxCD.

The extracellular sulfur was further characterized via SEM, XRD, and Raman spectroscopy. Morphological observations under SEM showed sulfur particles in two shapes: rod and globule. The length of rod-shaped sulfur can reach up to 30 μm, and some rod-shaped sulfur has short branches. The diameter of sulfur globules ranges from several hundred nanometers to 8 um, mostly 2–3 μm (Figure 3a,b). The XRD patterns of biogenic S^0^ matched well with the diffraction patterns of α-S_8_ in the database (Figure 3c). In addition, the Raman spectroscopy of S^0^ particles also showed the presence of α-S_8_ compared with a commercial S^0^ standard (Figure 3d). These results suggest that the elemental sulfur produced by strain XGS-01^T^ are aggregates of nanocrystalline α-S_8_, which is similar to most currently reported extracellular sulfur types, such as *Allochromatium vinosum*, *Chlorobaculum tempidum*, and *Sulfurimonas hydrogeniphila* [27,28,29]. The produced a-S_8_ has been found in very diverse environments, such as marine sediments, water columns, sulfidic caves, and hydrothermal vents, and the pattern is the most thermodynamically stable form of elemental sulfur at ambient temperature and pressure [23,24,25]. The crystalline sulfur could serve as an important intermediate in the biogeochemical sulfur cycle and could be further consumed by a wide diversity of microorganisms, such as sulfur oxidizers, sulfur reducers, or sulfur disproportionators.

## 4. Conclusions

A novel *Thiomicrorhabuds* species was characterized from coastal sediments, which contributed to our understanding of the ecological role and environmental adaptation of *Thiomicrorhabuds* bacteria in marine environments. The result of comparisons of the complete 16S rRNA gene sequence showed that the sequence of strain XGS-01^T^ was at least 2.2% different from other *Thiomicrorhabdus* members described previously. Pairwise ANI values between strain XGS-01^T^ and other species of the genus *Thiomicrorhabdus* were 70.0–72.0%. According to the generally recognized criteria for delineating bacterial species, strains with a 16S rRNA gene sequence dissimilarity of greater than 2% and ANI values of less than 95–96% are considered to belong to separate species. Thus, strain XGS-01^T^ does not belong to any presently described species. Although strain XGS-01^T^ shared the highest 16S rRNA gene sequence similarity to *T. hydrogeniphila* MAS2^T^, many of the physiological characteristics of strain XGS-01^T^ are different from strain MAS2^T^ (Table 1). The growth conditions and maximum growth rate differed between the two strains. The utilization patterns of electron donors were also different in that strain XGS-01^T^ could not use hydrogen as sole energy source, whereas strain MAS2^T^ could. In addition, strain XGS-01^T^ is able to utilize diverse inorganic nitrogen and organic nitrogen, whereas strain MAS2^T^ only uses ammonium as a nitrogen source. There are also significant differences between strain XGS-01^T^ and MAS2^T^ in the composition of fatty acids. These results clearly indicate that strain XGS-01^T^ can be differentiated from *T. hydrogeniphila* MAS2^T^ at the species level. Correspondingly, genomic analysis confirmed that strain XGS-01^T^ performed thiosulfate oxidation through a *sox* multienzyme complex and harbored *fcc* and *sqr* genes to catalyze sulfide oxidation. Comparative genomics further revealed that carbon fixation via the oxidation of reduced sulfur compounds coupled with oxygen reduction is conserved metabolic pathways among members of genus *Thiomicrohabdus*, indicating they play an important role in the sulfur oxidation process in aerobic marine environments. In addition, our study used strain XGS-01^T^ as the model with which to first characterize the extracellular sulfur produced by genus *Thiomicrorhabdus*. The produced a-S_8_, as the most thermodynamically stable form of elemental sulfur, could serve as an important intermediate in the biogeochemical sulfur cycle and could be further consumed by a wide diversity of microorganisms. Our results provide new insight into the potential ecological significance of extracellular S^0^ produced by *Thiomicrorhabdus* in coastal marine ecosystems.

### Description of Thiomicrorhabdus lithotrophica sp. nov

*Thiomicrorhabdus lithotrophica* XGS-01^T^.nov. (li.tho.tro’phi.ca. Gr. n. lithos, stone; N.L. fem. adj. trophica (from Gr. fem.adj. trophik), nursing, tending, or feeding; N.L. fem. adj. *lithotrophica*, inorganic-substrate-consuming.)

Cells are Gram-stain negative, rod-shaped (1.6–1.8 µm × 0.7–0.8 µm), motile, and have non-flagella. They are grown at 4–45 °C (optimum temperature, 30 °C), pH 5.0–9.0 (optimum pH 7.0), and 170–850 mM NaCl (optimum 510 mM). Strictly aerobic chemoautotrophic growth occurred with reduced sulfur compounds such as sulfide, thiosulfate, tetrasulfate, and elemental sulfur as electron donors and molecular oxygen as the only electron acceptor. Ammonium, nitrogen, nitrate, and aminotriacetic acid are utilized as nitrogen sources. It is sensitive to chloramphenicol (0.1 g/L), kanamycin (0.1 g/L), ampicillin (0.1 g/L), streptomycin (0.1 g/L), and vancomycin (0.1 g/L).

Type strain XGS-01^T^ (=MCCC 1A18865^T^) was isolated from the coastal sediments of Xiamen, China. The GenBank accession number for the genome sequence of *Thiomicrorhabdus lithotrophica* XGS-01^T^ is CP102381, and the 16S rRNA sequence number is OQ978225.

## Figures and Tables

**Figure 1 microorganisms-11-02569-f001:**
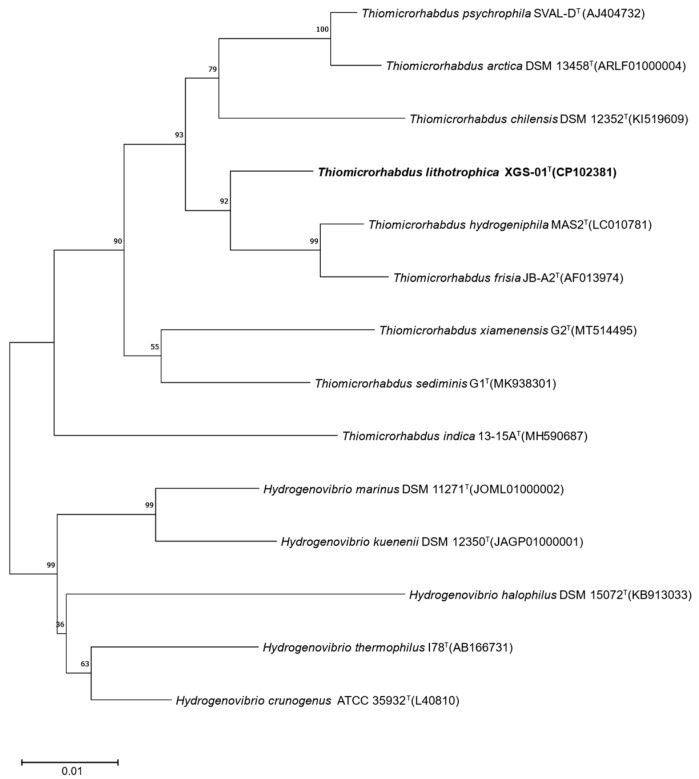
Neighbor-joining phylogenetic tree based on 16S rRNA gene sequences showing the relationship between strain XGS-01^T^ and members of the genus *Thiomicrorhabdus*. Bootstrap values (>50%) based on 1000 replicates are shown at branch nodes. Bar, 0.01 substitutions per nucleotide position.

**Figure 2 microorganisms-11-02569-f002:**
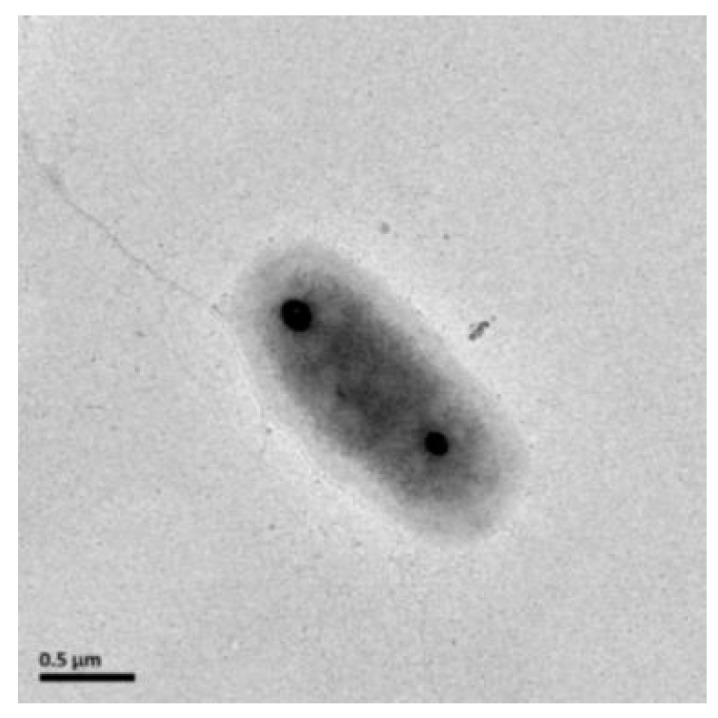
Transmission electron microscopy of a cell of strain XGS-01^T^.

**Figure 3 microorganisms-11-02569-f003:**
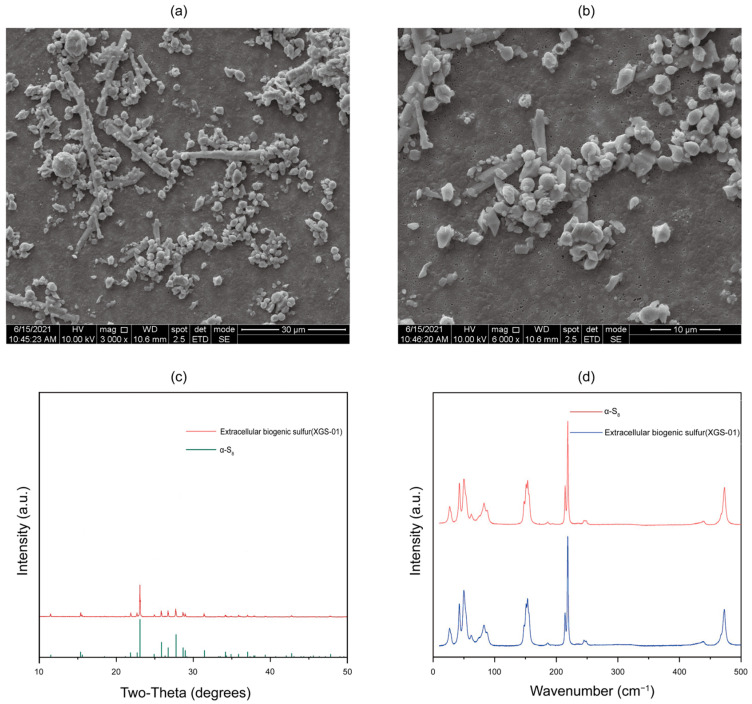
The morphology and structure of extracellular sulfur. (**a**,**b**) Extracellular sulfur morphology under scanning electron microscopy; (**c**) XRD pattern of extracellular sulfur; (**d**) Raman spectroscopy of extracellular sulfur.

**Table 1 microorganisms-11-02569-t001:** Comparison of characteristics of strain XGS-01^T^ and other species within the genus *Thiomicrorhabdus*. Strains: 1, *T. lithotrophica* XGS-01^T^; 2, *T. hydrogeniphila* MAS2^T^; 3, *T. frisia* JB-A2^T^; 4, *T. psychrophila* SVAL-D^T^; 5, *T. arctica* DSM 13458^T^. +, positive; −, negative.

Characteristic	1	2	3	4	5
Shape	Rod	Rod	Rod	Rod	Rod
G+C content (mol%)	39.1	39.6	39.6	42.5	42.2
Maximum growth rate (h^−1^)	0.15	0.40	0.45	0.20	0.14
Temperature range (°C)	4.0–45.0	2.0–40.0	3.5–39.0	−2.0–20.8	−2.0–20.8
Optimal temperature (°C)	30.0	30.0	25.0–32.0	14.6–15.4	11.5–13.2
Optimal Na^+^ concentration (mM)	510	270	470	250	250
pH range	5.0–9.0	5.0–8.0	4.2–8.5	6.5–9.0	6.5–9.0
Optimal pH	7.0	6.0	6.5	7.5–8.5	7.3–8.0
Electron donor:					
Thiosulfate	+	+	+	+	+
Sulfide	+	+	+	+	+
Elemental sulfur	+	+	+	+	+
Sulfite	−	−	−	−	−
Tetrathionate	+	+	+	+	+
Hydrogen	−	+	−	−	−

**Table 2 microorganisms-11-02569-t002:** Cellular fatty acid composition of strain XGS-01^T^, *T. hydrogeniphila* MAS2^T^, *T. sediminis* G1^T^, and *T. xiamenensis* G2^T^. Strain: 1, XGS-01^T^; 2, *T. hydrogeniphila* MAS2^T^; 3, *T. sediminis* G1^T^; 4, *T. xiamenensis* G2^T^. Values are percentages of total fatty acids. ND, Not detected.

Fatty Acid	1	2	3	4
Saturated:				
C_9:0_	0.23	ND	1.11	2.85
C_11:0_	0.06	ND	ND	1.01
C_12:0_	0.43	1.00	1.63	1.08
C_14:0_	1.38	0.80	3.61	2.90
C_16:0_	15.97	16.90	27.03	25.65
C_17:0_	ND	ND	1.34	1.18
C_18:0_	5.91	1.90	33.79	29.42
C_19:0_	0.17	ND	ND	2.24
Unsaturated:				
C_16:1_	6.90	44.40	3.60	3.17
C_17:1_	0.81	ND	4.87	3.17
C_18:1_	50.14	30.40	9.21	7.47
C_18:2_	0.56	ND	3.60	2.59
C_18:3_	0.42	ND	1.32	1.27
Hydroxy:				
C_8:0_ 3OH	0.25	ND	0.68	0.57
C_12:0_ 3OH	5.01	ND	1.76	3.03
C_18:1_ 2OH	0.49	ND	4.25	ND

**Table 3 microorganisms-11-02569-t003:** Genome comparison of five strains. Strain 1, *T. lithotrophica* XGS-01^T^; 2, *T. frisia* JB-A2^T^; 3, *T. arctica* DSM 13458^T^; 4, *T. chilensis* DSM 12352^T^; 5, *T.* sp. Milos-T2^T^. +, positive; −, negative.

Characteristic	1	2	3	4	5
Genome size (Mb)	2.5	2.7	2.6	2.4	2.6
GC content (%)	39.1	39.6	42.2	49.9	38.0
ANI	−	71.1	70.1	70.4	71.5
Gene count	2292	2526	2337	2285	2382
rRNA operons	9	3	3	2	3
tRNA count	47	45	45	43	46
Sox	ABCDXYZ	ABCDXYZ	ABCDXYZ	ABCDXYZ	ABCDXYZ
Fcc	+	+	−	+	+
Sqr	+	+	+	+	+
TCA Cycle	+	+	+	+	+
Pentose phosphate pathway	+	+	+	+	+
Glycolysis/Gluconeogenesis	+	+	+	+	+
Carboxysome	+	+	+	+	+
Carbonic anhydrase	1α-class, 1γ-class	1α-class, 1γ-class, 1β-class	1α-class, 1γ-class, 1β-class	1α-class, 2γ-class	1α-class, 1γ-class, 1β-class
Nitrate reductase	−	+	−	−	−
Nitrite reductase	−	+	−	+	−

## Data Availability

The GenBank accession numbers for the 16S rRNA gene sequence and the complete genome sequence of *Thiomicrorhabdus lithotrophica* XGS-01^T^ are OQ978225 and CP102381, respectively.

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
