# Peer review of "Physiological and Genomic Characterization of a Novel Obligately Chemolithoautotrophic, Sulfur-Oxidizing Bacterium of Genus Thiomicrorhabdus Isolated from a Coastal Sediment"

_microorganisms, 2023, doi:10.3390/microorganisms11102569_

Round 1
Reviewer 1 Report
I have few minor comments below:
1. Author should add line number for article.
2. Please add the aim of your study in introduction. Why the authors isolated the designated strain ?
3. Add the strain deposition number in abstract. Check the format for novel species deposition number in other articles. Cultures of novel strain have to deposited in two internationally recognized culture collections and obtained official certificate of deposit to be submitted with the manuscript. I found only one. Please, check a previously published taxonomic article for other requirements.
4. 2.3.1 16S rRNA gene phylogeny (correct this).
5. In 2.3.16. S rRNA gene phylogeny, add abbreviations for tree names like neighbor joining (NJ), maximum-likelihood (ML). After this sentence, only use abbreviation, don’t use the full name. Correct in other parts on manuscript that I have didn’t mentioned.
6. Correct maximum likelihood tree as maximum-likelihood (ML).
7. Add maximum parsimony (MP) tree instead of minimum evolution tree.
8. Add citation for Kimura two parameter model.
9. Transmission electron microscopy (TEM).
10. Gram-staining kit
11. As I already mentioned above please add maximum-parsimony tree which is more reliable instead of minimum evolution. Please add MP tree and compare the results with NJ and ML in result portion.
12. Use one numerical or unit in the whole manuscript.
13. Author mentioned that the designated strain was non-flagellated but via TEM picture I found there is flagella from left side. Please clarify this. Other members of Thiomicrorhabdus don’t have flagella so it doesn’t mean all members should not have flagella. There are many flagellated and non- flagellated members in the genus Lysobacter or other genus.
14. I think font for the following sentence should be corrected:
carboxyl oxygenase (RubisCO) and ribulose diphosphate (RuBP)
15.The author should improve the conclusion part. Clearly mention all the significance of the study.
16. Please clarify the difference between the novel strain and its reference strain on the basis of phenotypic and genotypic information.
Reviewer 2 Report
The study of Gao et al. characterized a novel species of belong to genus Thiomicrorhabdus using a polyphasic taxonomic approach from the costal line of Xiamen, China. In addition, we performed comparative genomic analyses among members of the genus Thiomicrorhabdus, to gain insights into the environmental adaptation mechanisms to elucidate their ecological roles and environmental adaptations.
The study is interested from the point of microorganisms. However, the ms need several improvements to be accepted.
It sems that the MS was submitted in a rush without proof reading or and correct formatting.
The manuscript needs reformatting, such as the abbreviation could be mentioned after first mention not before the abstract.
The references style is not fit the journal guidelines.
I’m confusing in the results section we have Fig 1 in the text but not in figures also there is fig S1 and Fig S2 in the text which duplicated in the figures.
The figure caption could write below the figure not above.
The discussion in several parts is insufficient and must be supported with recent knowledge.
The language needs native proof-reading where there are a several a long-sentences (without lines number how can I indicate it to be corrected). In addition several typo-grammatical errors (Transporting correct to transportation).
The language needs native proof-reading
